# Cloning and Characterization of a Novel Endo-Type Metal-Independent Alginate Lyase from the Marine Bacteria *Vibrio* sp. Ni1

**DOI:** 10.3390/md20080479

**Published:** 2022-07-26

**Authors:** Li Sha, Minghai Huang, Xiaonan Huang, Yongtong Huang, Ensi Shao, Xiong Guan, Zhipeng Huang

**Affiliations:** 1College of Life Sciences, Fujian Agriculture and Forestry University, Fuzhou 350002, China; shal@fafu.edu.cn (L.S.); minghai.huang@snibe.cn (M.H.); huangyongtong2022@163.com (Y.H.); 2Key Laboratory of Biopesticide and Chemical Biology, Ministry of Education, Fujian Agriculture and Forestry University, Fuzhou 350002, China; guanxfafu@126.com; 3Ministerial and Provincial Joint Innovation Centre for Safety Production of Cross-Strait Crops, Fujian Agriculture and Forestry University, Fuzhou 350002, China; 4Fuzhou Ocean and Fisheries Technology Center, Fuzhou 350007, China; hxn36@163.com; 5National Engineering Research Center of Juncao Technology, Fujian Agriculture and Forestry University, Fuzhou 350002, China; es776@fafu.edu.cn

**Keywords:** alginate lyase, alginate, oligosaccharide, PL7, product distribution, *Vibrio*

## Abstract

The applications of alginate lyase are diverse, but efficient commercial enzymes are still unavailable. In this study, a novel alginate lyase with high activity was obtained from the marine bacteria *Vibrio* sp. Ni1. The ORF of the *alg*B gene has 1824 bp, encoding 607 amino acids. Homology analysis shows that AlgB belongs to the PL7 family. There are two catalytic domains with the typical region of QIH found in AlgB. The purified recombinant enzyme of AlgB shows highest activity at 35 °C, pH 8.0, and 50 mmol/L Tris-HCl without any metal ions. Only K^+^ slightly enhances the activity, while Fe^2+^ and Cu^2+^ strongly inhibit the activity. The AlgB preferred polyM as substrate. The end products of enzymatic mixture are DP2 and DP3, without any metal ion to assist them. This enzyme has good industrial application prospects.

## 1. Introduction

Alginate is a linear polysaccharide in which β-d-mannuronate (M) and α-l-guluronate (G) are randomly linked by (1–4)-covalent bonds [1]. It is a natural gelling polysaccharide present in cell walls and intracellular material of the brown seaweeds, and is also produced by two families of heterotrophic, Gram-negative bacteria, namely the Pseudomonadaceae and the Azotobacteriaceae [2]. Alginate lyases (EC 4.2.2.3/4.2.2.11) can catalyze the degradation of alginate by cleaving the glycosidic bond through a β-elimination reaction, generating an unsaturated oligomer with 4-deoxy-l-erythro-hex-4-enepyranosyluronate at the nonreducing end [3,4]. In a natural environment, alginate lyases play an important role to recycle alginate. 

Original research on alginate lyase was from *Pseudomonas aeruginosa* [5,6,7]. Currently, alginate lyase have been investigated in many species, including marine algae, marine mollusks, fungi, viruses and a wide range of marine and terrestrial bacteria [8,9,10,11,12,13]. Based on the homology of the primary sequences, alginate lyases are classified into twelve polysaccharide lyase families including PL5, PL6, PL7, PL14, PL15, PL17, PL18, PL31, PL32, PL34, PL36, and PL39, most belong to PL5 and PL7 [8,14,15].

The products produced by alginate lyases, alginate oligosaccharides, have diverse bioactivities, such as anti-oxidant [16], anti-inflammatory [17], anti-tumor proliferation [18,19], anti-apoptotic [20], and have potential value in food and pharmaceutical industries [21,22]. Since alginate lyases are effective tools to prepare alginate oligosaccharides, the attention on their research is increasing. 

In the present study, a novel alginate gene was cloned from *Vibrio* sp. Ni1 and over-expressed in *Escherichia coli*. The purified recombinant enzyme has high activity toward alginate, poly M and poly G at high temperature and pH, independent of ions. The properties of the recombinant enzyme and its products were analyzed.

## 2. Results and Discussion

### 2.1. The Analysis of Alginate-Degrading Activity of Vibrio sp. Ni1

After growth on the alginate plate for 48 h, three colonies of *Vibrio* sp. Ni1 and their alginate-degraded circles were observed and measured (Figure 1 and Table 1). The results of plate assay showed that the central bright yellow zones were formed by iodate was reduced to I_2._ It suggested that *Vibrio* sp. Ni1 can produce alginate lyase to degrade algin and yield the unsaturated uronic acid with reducibility. The diameter of alginate-degraded zones were more than four times greater than that of the colonies. The result showed that *Vibrio* sp. Ni1 has good ability to degrade alginate.

### 2.2. The LC-MS/MS Analysis of Crude Alginate Lyase

The crude alginate lyase solution was purified by 70% (NH_4_)_2_SO_4_ precipitation. The majority of the extracellular proteins were exhibited in the range of 20–75 kDa in SDS-PAGE gel (Figure 2). The protein bands in lane 1 were cut-out to extract the proteins, then the proteins were analyzed by LC-MS/MS. There were 147 peptides detected and 6 peptides (Table 2) were discovered to match with one candidate alginate lyase gene. According to the homology of AIY22661.1 and WP_118120558.1, and peptide locations, No. 3 and 4 peptide sequences were chosen to design a pair of degenerate primers named algB-F0/R0 to amplify the partial *alg*B gene.

### 2.3. The Characterization of the Full Length algB Gene

Based on the sequencing result of the partial *alg*B gene, using SiteFinding-PCR technology, a series of primers were designed to obtain the full length *alg*B gene. Finally, a pair of primers named *alg*B-QF/QR were designed to amplify the full length of *alg*B gene (Figure 3). It consisted of 1824 bp and encodes 607 amino acids with a sequence of 18 amino acids at the *N*-terminus that was predicted to be signal peptide by online software Signal P5.0. The AlgB protein had a calculated molecular weight 67.7 kDa and pI 4.34. It was subjected to Blast analysis in the NCBI database. The DNA sequence displays high similarity with only one partial genome of *Vibrio penaeicida* (AP025146.1, 93.81% identity of 86% length). The amino acid sequence displays high similarity of 93.87% identity with three alginate lyases (WP_148112135.1 from *Vibrio penaeicida*, WP_224056095.1 from *Vibrio penaeicida*, WP_224915578.1 from *Vibrio alginolyticus*). These three alginate lyases have not been characterized. Other alginate lyases have about 70–80% identities, such as AIY22661.1 from *Vibrio* sp. W13, 81.13%; and WP_118120558.1 from *Vibrio* sp. Dhg, 77.86% (Figure 4a). The *alg*B sequence has been submitted to NCBI (GenBank accession No. MZ680650).

AlgB is classified in the polysaccharide lyase family 7 (PL7). Conserved domain analysis revealed two structural domains in AlgB (Figure 4b). Each structural domain has a predicted catalytic domain. Alginate lyases in PL7 contain three highly conserved motifs, R*E*R, Q(I/V)H, Y*KAG*Y*Q, which form the active site and play an important role in substrate binding and catalysis [23,24]. These conserved motifs also exist in each of the two structural domains in AlgB (RSEVR, QIH, YFKAGVYNQ; RSELR, QIH, YFKAGIYPH). The predicted three-dimensional structure of AlgB was generated by LOMETS (Figure 4c). The typical protein structure of PL7 is a single domain β- jelly roll [3,23]. The first structural domain is residues 78–310, the second structural domain is residues 319–606. These two structural domains connect closely with an octapeptide linking region. It is rare that PL7 alginate lyases have two catalytic domains [25,26].

### 2.4. Purification of the Recombinant Enzyme AlgB

To characterize the recombinant enzyme, the *alg*B gene was cloned into the expression vector pET32a (+) and transformed to *E. coli* BL21. The fusion proteins of AlgB were soluble when expressed in host cells and purified by nickel affinity chromatography. Since the mass of the fusion tag protein (TrxA) is approximately 20 kDa, the recombinant AlgB is about 88 kDa. The SDS-PAGE analysis is consistent with the prediction (Figure 5). The activity of purified AlgB was measured (Figure 6). Along with the increase in time, the absorption at λ235 was enhanced rapidly within 1 min and, subsequently, increased slowly. It suggested that AlgB shows the activity of alginate lyase.

### 2.5. Characteristics of the Recombinant Enzyme AlgB

The optimal pH for the recombinant enzyme AlgB activity was 8.0 and the highest activity was 979 U/mL (Figure 7). More than 50% of the maximum activity was measured at a wide pH range 5.0–10.0 and the activity was stable at pH range of 6.0–10.0. The activity of AlgB sharply decreased when pH lower than 5.0. Beyond Tris-HCl buffer capacity, the activity was not measured on the condition of pH higher than 10.0. The results suggest that the AlgB has good activity at basic pH conditions. The characteristic of some alginate lyases with similar molecular mass from *Vibrio* strains are summarized (Table 3). These alginate lyase showed optimal pH ≥ 7.0, the highest pH 8.9 [25,26,27,28,29,30,31,32]. 

The optimal temperature for AlgB was 35 °C and the highest activity was 1059 U/mL (Figure 8a). More than 50% of the maximum activity was measured at a wide temperature range 20–60 °C. The activity of AlgB has sharply decreased over 60 °C. Even at 4 °C, the activity of AlgB was measured 35% maximum. The enzyme retained more than 50% of the highest activity after being incubated at 30 °C for 36 h, and the enzyme retained more than 40% of the highest activity after being incubated at 40 °C for 24 h (Figure 8b). The results show that the activity of AlgB is stable at 30–40 °C. The optimal temperature of AlgB was similar to Aly-IV from *Vibrio* sp. QD-5 [31].

The activity of AlgB was inhibited by 10 mmol/L Mg^2+^, Ca^2+^, Mn^2+^, Co^2+^, Ni^2+^, Zn^2+^; furthermore, strongly inhibited by Fe^2+^ and Cu^2+^. Among the metal ions, only K^+^ displayed slight activation of enzyme activity (Figure 9a). The enzyme retained more than 50% of the highest activity in 1–2 mol/L NaCl (Figure 9b). Enhancing the concentration of NaCl to more than 3 mol/L, the activity sharply decreases. These results suggested that AlgB has a good tolerance to NaCl. Previous investigations report that many metal ions can increase the enzyme activity, such as Mg^2+^ and Ca^2+^ (Table 3). The effect of metal ion on AlgB was similar to Aly-IV except for Mg^2+^ (Table 3). With the same concentration, Mg^2+^ strongly activates Aly-IV but slightly inhibits AlgB.

The kinetic parameters of AlgB were determined at the optimal reaction conditions. The specific activity of AlgB towards sodium alginate was 2118 U/mg, the K_m_ value of AlgB was 8.22 mg/mL, the value of V_max_ was 2.6 mg/min, and the value of K*_cat_* was 26 min^−1^ at pH 8.0, 35 °C (Figure 10).

### 2.6. Substrate Specificity of AlgB

To investigate their substrate specificity, sodium alginate, polyM and polyG were used as substrate for the recombinant enzyme AlgB. Compared to sodium alginate, the relative activity of AlgB to polyM was 16% higher, while polyG was 14% lower at the same reaction conditions (Figure 11). This illustrates that AlgB degrades M-M links more efficiently. Both AlyA and AlyA-OU2 which were favorably to degrade polyM in activity assays (Table 3). The structure of these two alginate lyases are similar to that of AlgB. There is a double region of QIH within them which determines the enzyme has polyM specificity. Some alginate lyases from *Vibrio* strains were reported sodium alginate-preferred, such as the alginate lyase from *Vibrio* sp. YWA, Aly-IV from *Vibrio* sp. QD-5 and Alg7A from *Vibrio* sp. W13. A few polyG-preferred alginate lyases were discovered, such as AlyB and AlgNJ04. It can be inferred that the QIH region is not absolutely determined by the substrate specificity. 

### 2.7. Analysis of AlgB Degradation Products

Degradation products of AlgB activity were analyzed by TLC and ESI-Q-TOF. The result of TLC showed that the distribution of the end products from the sodium alginate were separated to two different regions with different colors (Figure 12). The negative ESI-Q-TOF results were similar to that of TLC (Figure 13). The main products of degrading sodium alginate, polyM, and polyG were similar and consist of DP2 and DP3. It suggests that the AlgB belongs to endo- alginate lyase family. These results suggest that dimers and trimers are the primary end products of AlgB activity.

The end products from AlgB degradation are similar to Aly-IV (Table 3). It is different from most other alginate lyases currently reported. The degradation mode of AlyA is mainly DP1–4; that of AlyA-OU02 is mainly DP2–4; Algb and AlgNJ04 are mainly DP2–5; and Alg7A is mainly DP2–6 (Table 3). 

The process of the recombinant enzyme AlgB was monitored in a viscosity assay (Figure 14). The viscosity of the sodium alginate solution rapidly decreased in the first 5 min, then the viscosity slowly decreased in the following 25 min. The variation trend of AlgB degradation resembled to Aly-IV [31]. 

## 3. Materials and Methods

### 3.1. Bacteria and Substrate

The strain of *Vibrio* sp. Ni1 was isolated from the sludge of the Minjiang river estuary, located in Fujian province, China. This strain was kept in our lab and China Center for Type Culture Collection (CCTCC No. M2018916). *E. coli* BL21(DE3) was used as an expression host. The sodium alginate donated by Haizhilin Biotechnology Development Co. Ltd. (Qingdao, China) was used as the substrate to test alginate lyase activity. The viscosity of this sodium alginate sample was 1250 mPa.s. The sodium alginate used to prepare medium was from Solarbio (Beijing, China).

### 3.2. Screening of Alginate-Degrading Activity

A 5 μL aliquot from a culture of *Vibrio* sp. Ni1 grown in liquid Luria-Bertani (LB) medium, was transferred to an alginate plate containing 1% (*w*/*v*) sodium alginate, 0.2% (*w*/*v*) glucose, 0.3% (*w*/*v*) NaCl, 0.2% (*w*/*v*) NaNO_3_, 0.05% (*w*/*v*) MgSO_4_·7H_2_O, 0.1% (*w*/*v*) K_2_HPO_4_, 0.01% (*w*/*v*) CaCl_2_, and 1% (*w*/*v*) agar, pH 7.5. The plate was incubated at 28 °C for 48 h. The plate was stained with 1 mL Lugol’s reagent (5% (*w*/*v*) I_2_, 10% (*w/v*) KI) for 1 min [33]. The plate was washed with deionized water and observed. 

### 3.3. Extraction of Crude Alginate Lyase

The cells of *Vibrio* sp. Ni1 were cultured in the liquid fermentation medium containing 1% (*w*/*v*) sodium alginate, 3% (*w*/*v*) NaCl, 0.2% (*w*/*v*) NH_4_Cl, 0.05% (*w*/*v*) MgSO_4_·7H_2_O, 0.1% (*w*/*v*) K_2_HPO_4_, and 0.01% (*w*/*v*) CaCl_2_, pH 7.5, at 28 °C for 24 h. The supernatant was harvested by centrifugation at 4 °C, 8000× *g*, for 10 min. Solid (NH_4_)_2_SO_4_ was gradually added to the supernatant to a final concentration of 70% (*w*/*v*), then stirred continuously at 4 °C for 12 h. The protein precipitant was collected by centrifugation (4 °C, 10,000× *g*, 30 min) then dissolved in 0.05 mol/L Tris-HCl (pH 7.5). The suspended solution was dialyzing in 0.05 mol/L Tris-HCl (pH 7.5) 12–15 h. The crude protein solution was obtained after centrifugation (4 °C, 10,000× *g*, 20 min) to get rid of insoluble impurity inside the dialysate, then stored at 4 °C. 

### 3.4. Identification of Alginate Lyase Fragments

The crude alginate lyase was separated by sodium dodecyl sulfate polyacrylamide gel electrophoresis (SDS-PAGE). The 10% acrylamide gel was stained with Coomassie blue and detained with 30% acetonitrile and 0.1 mol/L NH_4_HCO_3_. The gel slice was dried in a vacuum centrifuge. The proteins were reduced in-gel with dithiothreitol (10 mmol/L DTT and 100 mmol/L NH_4_HCO_3_) for 30 min at 56 °C, then alkylated with iodoacetamide (20 mmol/L IAA and 100 mmol/L NH_4_HCO_3_) in the dark at 25 °C for 30 min. The gel slice was briefly rinsed with 100 mmol/L NH_4_HCO_3_ and acetonitrile, followed by digestion with 12.5 ng/μL trypsin in 25 mmol/L NH_4_HCO_3_ overnight. The digested peptides were extracted three times with 60% (*v*/*v*) acetonitrile and 0.1% (*v*/*v*) trifluoroacetic acid. The extracts were pooled and dried completely in a vacuum centrifuge. The peptide mass and sequence were determined by Liquid Chromatography (LC)-Electrospray Ionization (ESI) Tandem mass spectrometry (MS/MS) in a Q Extractive mass spectrometer which was coupled to Easy nLC (Proxeon Biosystems, Thermo Fisher Scientific, Waltham, MA, USA). The MS data were analyzed using Max Quant (Version 1.6.4.0) by searching the data against the amino acid sequence of alginate lyases from *Vibrio*, and the intensity of sequenced peptide in the target protein was calculated.

### 3.5. Cloning and Sequence Analysis of the Alginate Lyase Gene

Based on the LC-MS/MS results, a pair of degenerate primers (algB-F0/R0) was designed to amplify fragments of the *alg*B gene. According to the sequencing result of the partial *algB* gene, a series of primers were designed to apply the SiteFinding-PCR method to extend the target gene [34,35]. The details of primers used in this experiment are listed in Table 4. Chromosome Walking Kit and Pfu DNA polymerase (TaKaRa, Osaka, Japan) were used in this study. Finally, a pair of primers named *alg*B-QF/R was designed to amplify the full-length gene. The expression recombinant plasmid was constructed with pET32a on sites of *BamH* I and *Xhol* I and transformed into *E. coli* BL21(DE3). 

### 3.6. Purification of the Alginate Lyase

The *E. coli* cells harboring the recombinant plasmid pET32a-*alg*B were cultured at 37 °C in liquid LB medium with 100 μg/mL ampicillin until OD_600_ reached about 0.6. Gene expression was induced with the addition of isopropyl-β-d-thiogalactoside (IPTG) to a final concentration of 0.8 mmol/L and then grown at 20 °C for 24 h. The cells were harvested by centrifugation 8000 r/min, 5 min, at 4 °C, then suspended in 0.05 mol/L Tris-HCl buffer (pH 7.4) to concentrate 1/10 volume. The cells were disrupted by ultrasonication. The lysate was harvested by centrifugation 10,000× *g*, 30 min, at 4 °C. The supernatant was filtered with 0.22 micrometer cellulose acetate membrane before passage through a Ni-NTA Sepharose column (GE, Boston, MA, USA). The column was washed with washing buffer (0.05 mol/L Tris-HCl buffer (pH 7.5), 0.5 mol/L NaCl, 20 mmol/L imidazole), and the recombinant alginate lyase AlgB was eluted with elution buffer (0.05 mol/L Tris-HCl buffer (pH 7.5), 0.5 mol/L NaCl, 0.3 mol/L imidazole). The eluate was dialyzed in 0.05 mol/L Tris-HCl buffer (pH 7.5) overnight and analyzed by 12% SDS-PAGE. The protein concentration was measured by the Bradford method.

### 3.7. Assay of the Recombinant Alginate Lyase Activity

The activity of the alginate lyase AlgB was measured in a mixture solution containing 1% (*w*/*v*) sodium alginate dissolved in 0.05 mol/L Tris-HCl buffer (pH 8.0) (0.9 mL) and enzyme solution (0.1 mL) at 40 °C for 30 min. Enzyme activity was calculated by constructing a standard curve with uronic acid under the same reaction conditions with DNS solution [36]. One unit (U) activity was defined as the amount of enzyme that generated 1 μg of uronic acid per min under standard assay conditions. In order to distinguish reaction products, the enzyme activity was also determined as the increase in the absorbance 235 nm to measure the amount of unsaturated uronic acid [15]. 

### 3.8. Characterization of the Recombinant Alginate Lyase Activity

The activity of AlgB at a specific pH range (3–10) and temperature range (4–90 °C) was determined by the enzyme assay described above using 1% alginate as substrate. The thermostability was evaluated by measuring the residual activity after different treatment time at different temperatures. Kinetic parameters of the enzyme towards alginate were calculated by the substrate at different concentrations (1–5 mg/mL). The enzyme activity was measured as described previously. 

The effect of metal ions on the activity of AlgB was investigated. The reaction mixtures were added with different metal ions (KCl, MgSO_4_, ZnSO_4_, FeSO_4_, CuSO_4_, MnSO_4_, CaCl_2_, CoCl_2_ and NiCl_2_) to a final concentration of 10 mmol/L. The parallel reaction without metal ions served as the control. The activity on different concentrations of NaCl (1–5 mol/L) was also measured.

### 3.9. Substrate Specificity of the Recombinant Alginate Lyase Activity

The purified AlgB was added to the reaction mixture containing 1 mg/mL alginate, polyG and polyM, respectively. The average molecular weights of both polyG and polyM (Shanghai Yuanye Bio-Technology Co. Ltd., Shanghai, China) were 6–8 kDa. 

### 3.10. Analysis of Reaction Mode and Products

The mixture of enzymatic products was purified by centrifugation (10,000 r/min, 30 min, 4 °C) after react 12 h at optimal conditions. The degraded products of AlgB were analyzed by thin-layer chromatography (TLC) (TLC Silica gel 60 F254, Merck KGaA, Darmstadt, Germany) [37]. The enzymatic products on silica gel plate were separated with a solvent solution of n-butyl alcohol/acetic acid/H_2_O (3:2:3) and visualized by heating 80 °C for 15 min after spraying a diphenylamine/aniline/phosphate (1:1:5) reagent. To further investigate the enzymatic products, ESI-QqQ-MS/MS was conducted. Five microliter samples were injected into a triple quadrupole tandem mass spectrometry (QqQ-MS/MS) (Aligent Technologies Inc., Palo Alto, CA, USA) after filtration with 0.22 µm cellulose acetate membrane. The products were detected in a negative ion mode using the following settings: pressure of the atomizer (N_2_), 50 psi; flow rate of the dryer (N_2_), 10 L/min; temperature of the dryer (N_2_), 350 °C; capillary voltage, 4000 V; scanning the mass range, 100–1000 *m*/*z* [25]. The viscosity of substrate was intermittently determined by viscometer (Shanghai Hengping No. NDJ-1, Shanghai, China) during reaction.

## 4. Conclusions

In this study, a novel alginate lyase of AlgB with high activity was obtained from *Vibrio* sp. Ni1. The purified recombinant enzyme of AlgB showed highest activity at 35 °C, pH 8.0, 50 mmol/L Tris-HCl without metal ions. The AlgB preferred polyM as substrate. The predominant end products of the enzymatic reaction are DP2 and DP3. 

Some alginate lyases can catalyze the degradation in extreme environments. The optimal pH of A1m [38] and AlgNJ-07 [39] is 9.0 from *Agarivorans* sp. JAM and *Serratia marcescens*, respectively. The optima pH and temperature of AlyPL6 [14] and AlgH [40] is 45 °C, pH 10. The highest optima temperature of a known alginate lyase from PL7 family is 70 °C found in NitAly [41] and A1-II [42]. As reported here, AlgB can degrade alginate at a wide range of temperatures (20–60 °C) and pH (6–10). These properties are good for industrial applications. 

Many organisms are reported to encode more than one alginate lyase gene, such as *Vibrio splendidus* OU02 [26,37], *Vibrio splendidus* 12B01 [25], *Sphingomonas* sp. A1 [42,43], *Flavobacterium* sp. UMI-01 [44], and *Aplysia Kurodai* [45]. As indicated in Figure 2, it is possible there are multiple alginate lyases in *Vibrio* sp. Ni. Work to identify additional alginate lyases from this strain are in progress.

## Figures and Tables

**Figure 1 marinedrugs-20-00479-f001:**
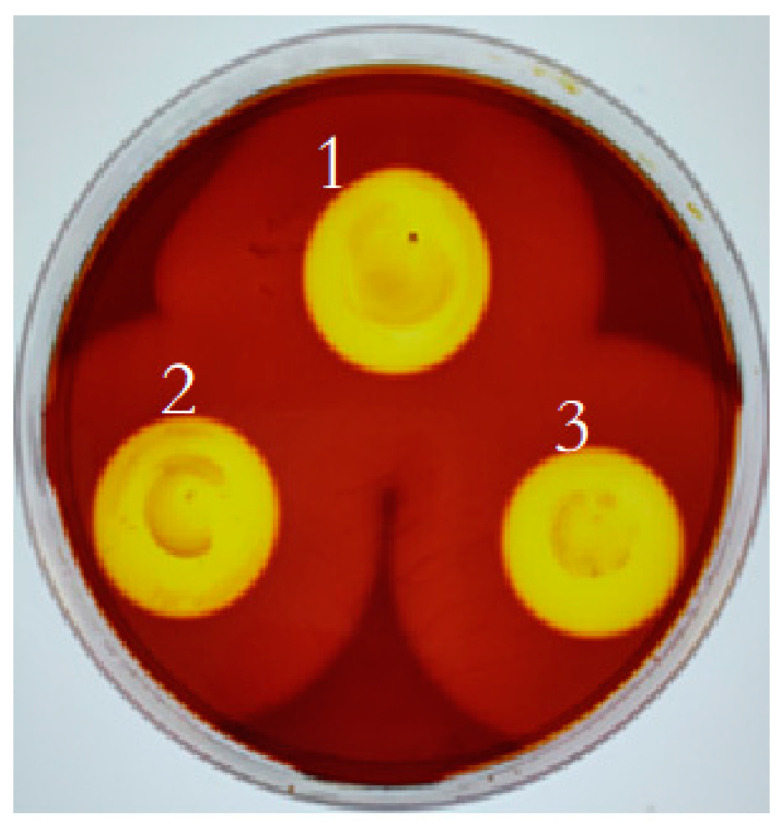
Halos on alginate plate formation by *Vibrio* sp. Ni1. Note: The different colors depend on the degree of degradation of algin.

**Figure 2 marinedrugs-20-00479-f002:**
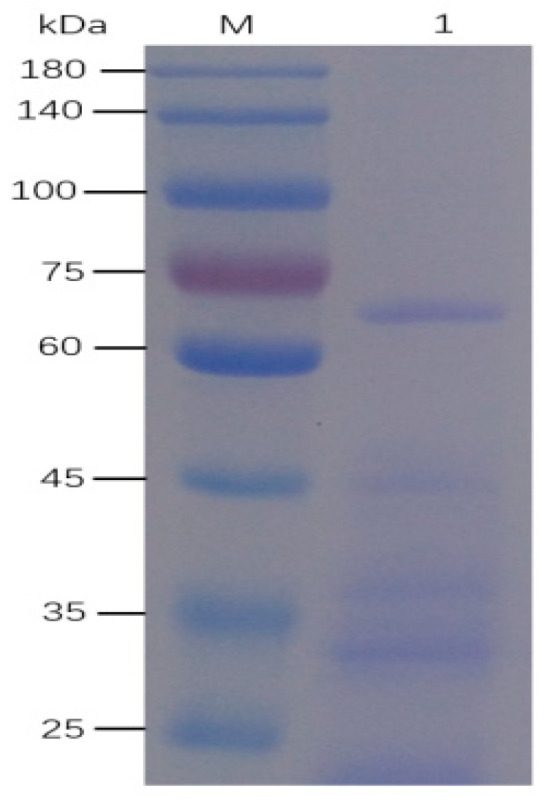
SDS-PAGE analysis of extracellular protein *Vibrio* sp. Ni1. Lane M: Protein Marker; Lane 1: the proteins purified by the 70% ammonium sulfate precipitation.

**Figure 3 marinedrugs-20-00479-f003:**
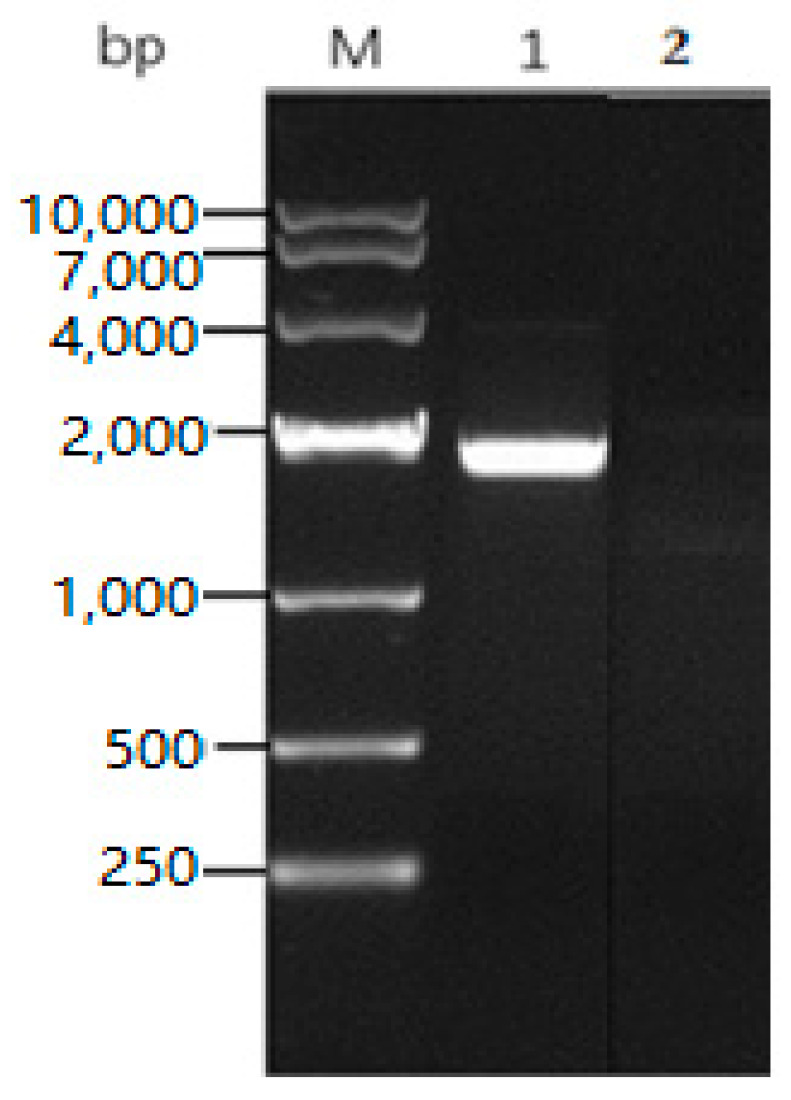
The PCR amplification of full-length *alg*B gene. Lane M: DNA Marker; Lane 1: The product of full-length *alg*B gene; Lane2: negative control.

**Figure 4 marinedrugs-20-00479-f004:**
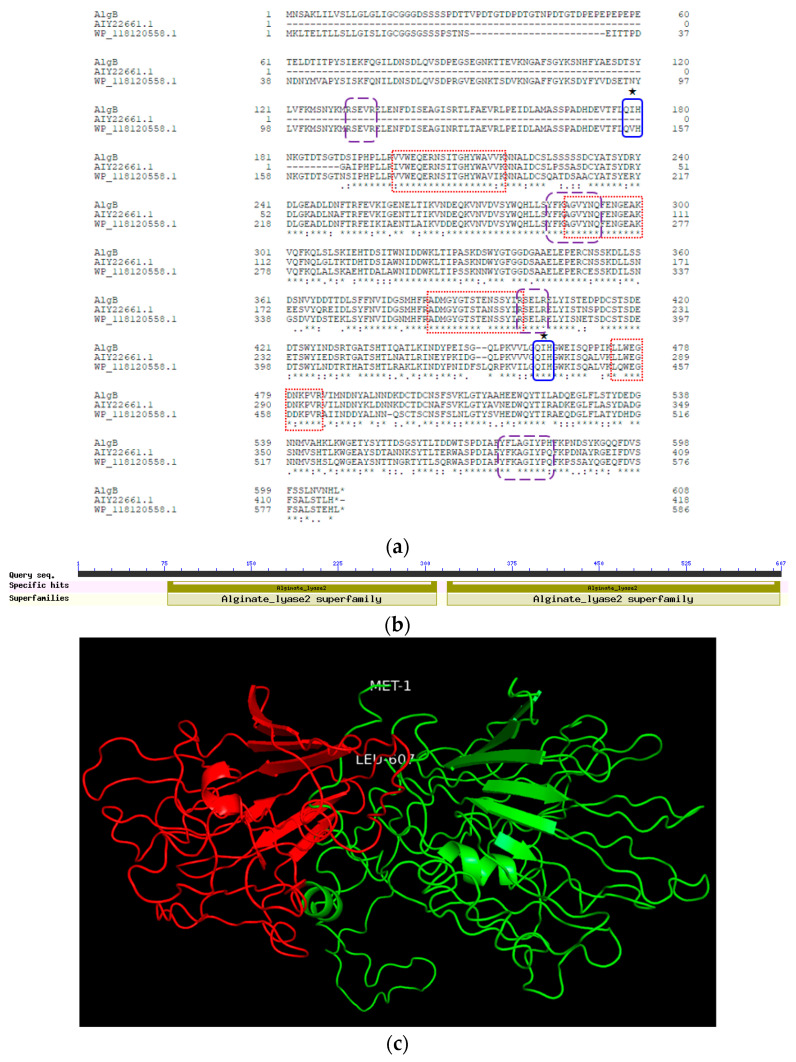
Sequence analysis and structure prediction of AlgB. (**a**) Multiple sequence alignment of AlgB and related alginate lyases (AIY22661.1 WP_118120558.1). The peptide sequences which match the results of LC-MS/MS are marked with red boxes. The conservative sequences of PL7 family catalytic domain are marked with blue or purple boxes. The predicted catalytic centers are marked with ★. (**b**) Functional domains of AlgB. (**c**) Schematic diagram of AlgB. The predicted structure of AlgB has two domains. The red domain (on left) is on N-terminal (residue 78–310), the green domain (on right) is on the C-terminal (residue 319–606). The N- and C-terminal amino acid residues are labeled Met1 and Leu607.

**Figure 5 marinedrugs-20-00479-f005:**
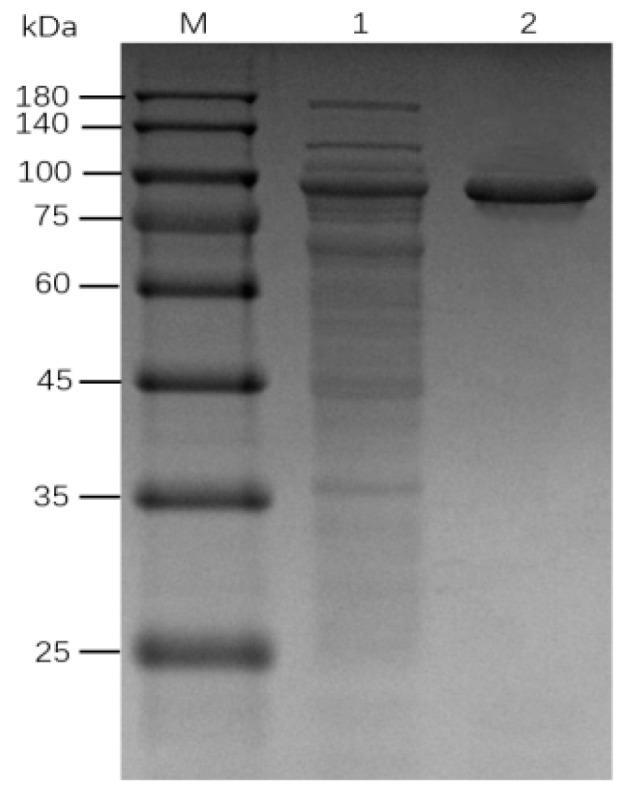
SDS-PAGE analysis of purified recombinant AlgB. Proteins were electrophoresed on 10% gel and stained with CCB G250. Lane M, protein marker; Lane1, total soluble proteins after cell rupture; Lane2, purified AlgB by Ni-column.

**Figure 6 marinedrugs-20-00479-f006:**
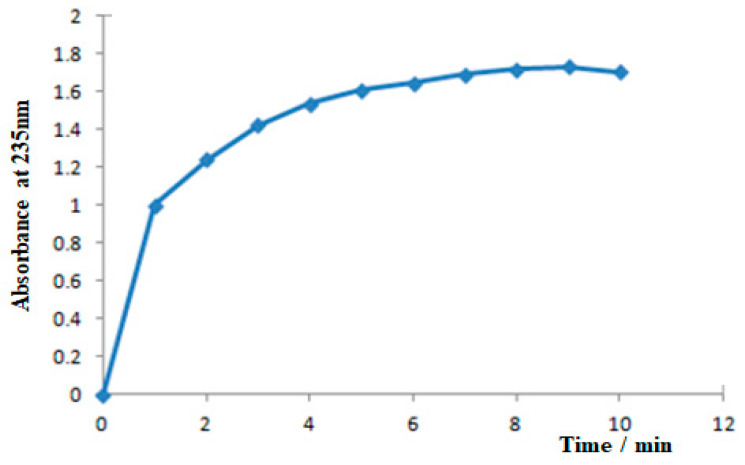
The changes of A235 during the process of degradation.

**Figure 7 marinedrugs-20-00479-f007:**
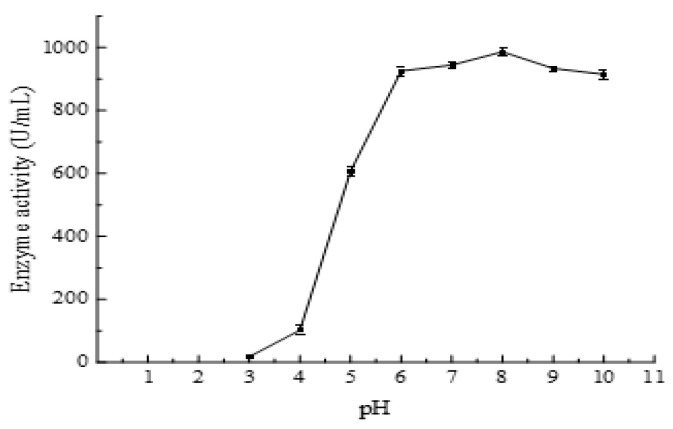
The effect of pH on the enzyme activity of AlgB. Reactions were conducted at 40 °C for 30 min in 50 mmol/L Tris-HCl buffer with a pH range 3–10. The data represent the mean of three experimental repeats with SD < 5%.

**Figure 8 marinedrugs-20-00479-f008:**
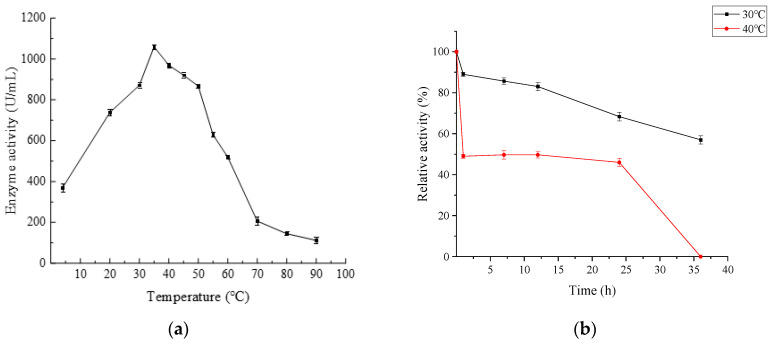
The effect of temperature on the activity of AlgB. (**a**) The optimal temperature of AlgB. Reactions were conducted in 50 mmol/L Tris-HCl buffer (pH 8.0) at 4–90 °C for 30 min. (**b**) Thermal stability of AlgB. Black square (■) represents at 30 °C and red dot (●) represents at 40 °C. The activity of AlgB measured in 35 °C and pH 8.0 was taken as 100%. The data represent the mean of three experimental replicates with SD < 5%.

**Figure 9 marinedrugs-20-00479-f009:**
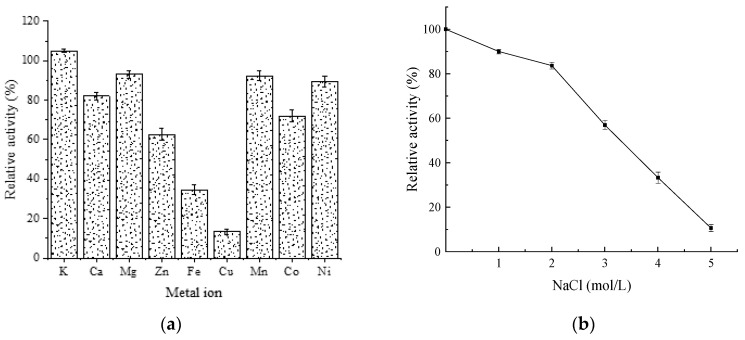
The effects of metal ions on the activity of AlgB. (**a**) Different metal ions on the activity of AlgB. The enzyme activity was measured at 35 °C and 50 mmol/L Tris-HCl buffer (pH 8.0) containing 10 mmol/L metal ions. (**b**) The effect of NaCl concentration on the activity of AlgB. The enzyme activity was measured at 35 °C and 50 mmol/L Tris-HCl buffer (pH 8.0) containing 1–5 mol/L NaCl. The activity of AlgB measured in 35 °C and 50 mmol/L Tris-HCl buffer (pH 8.0) without any metal ions was taken as 100%. The data represents the mean of three experimental repeats with SD ≤ 5%.

**Figure 10 marinedrugs-20-00479-f010:**
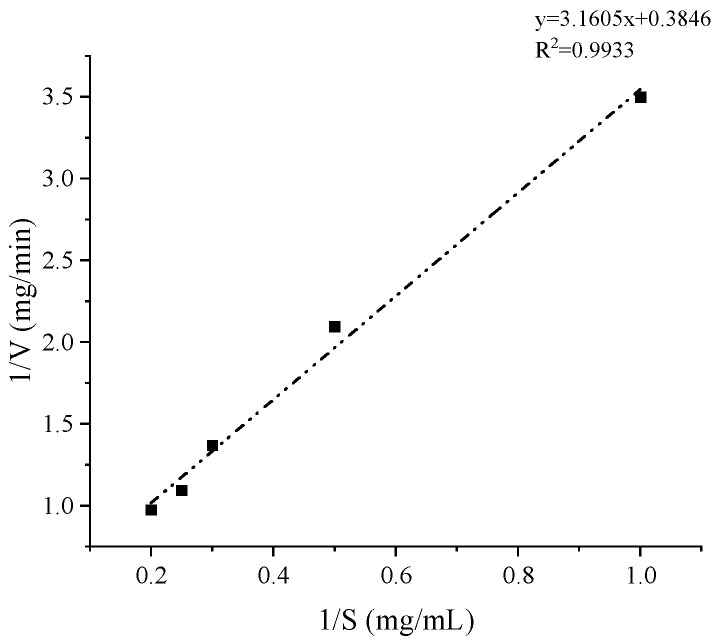
Lineweaver-Burke plots for the degradation of alginate by AlgB. Enzyme activity was determined at different alginate concentration of 1.0–5.0 mg/mL, 35 °C and 50 mmol/L Tris-HCl buffer (pH 8.0) according to Nelson method. The data represents the mean of three experimental repeats with SD ≤ 5%.

**Figure 11 marinedrugs-20-00479-f011:**
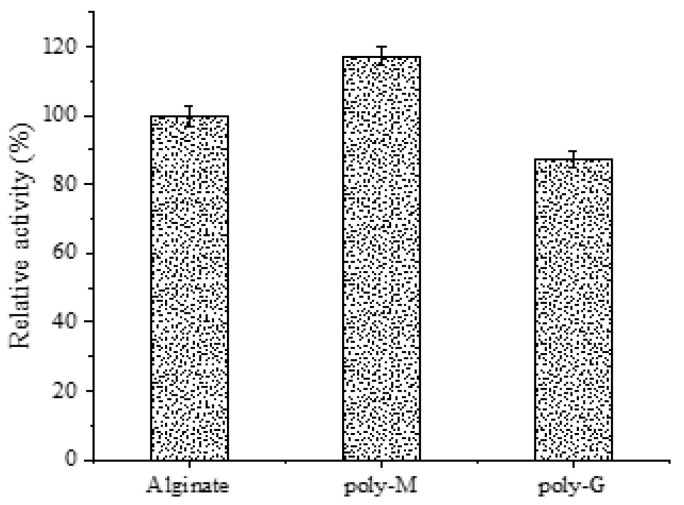
Substrate specificity of AlgB. The concentration of substrate was 1 mg/mL. The activity measured in 35 °C and 50 mmol/L Tris-HCl buffer (pH 8.0) using sodium alginate as substrate was taken as 100%. The data represent the mean of three experimental repeats with SD ≤ 5%.

**Figure 12 marinedrugs-20-00479-f012:**
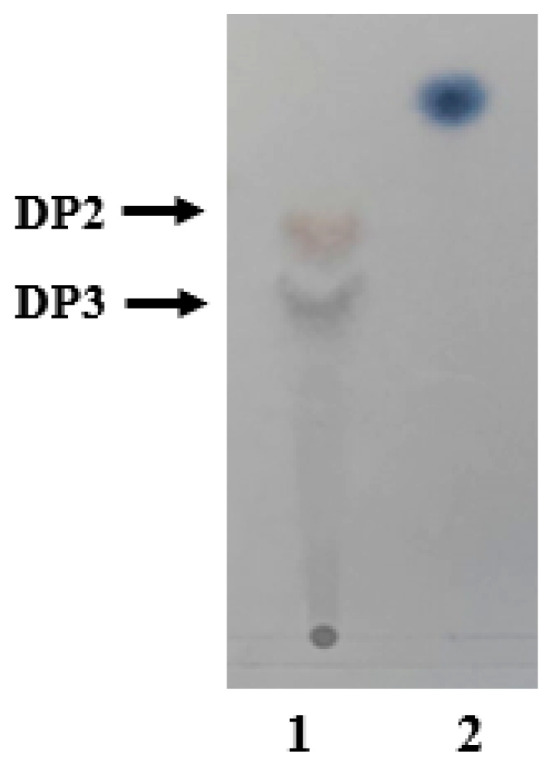
TLC analysis of degradation products of AlgB towards sodium alginate. Lane 1, the enzymatic products after reaction with AlgB for 12 h; Lane 2, mannuronic acid.

**Figure 13 marinedrugs-20-00479-f013:**
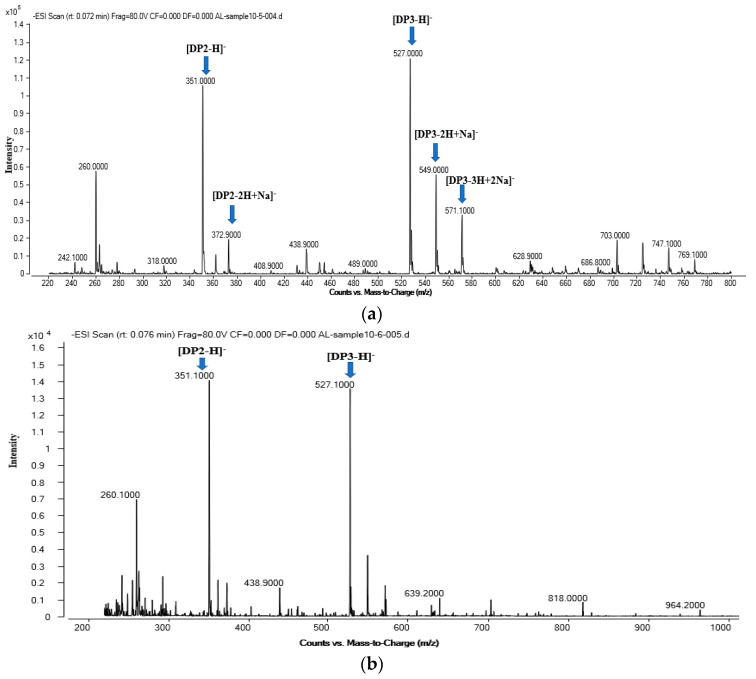
ESI−Q−TOF analyses of the end products of AlgB. The enzymatic products were harvested after AlgB incubation for 12 h. (**a**) sodium alginate, (**b**) polyM, and (**c**) polyG. DP indicates the degree of depolymerization of oligosaccharides released from the substrate.

**Figure 14 marinedrugs-20-00479-f014:**
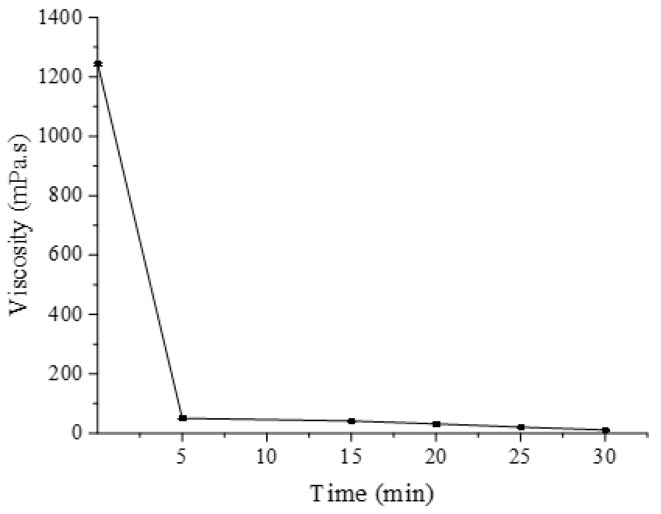
Viscosity variation during enzymatic degradation of alginate. The viscosity was measured in a 100 mL reaction system with 1% (*w*/*v*) sodium alginate, 50 mmol/L Tris-HCl buffer (pH 8.0) and RT. The data represents the mean of three repeats with SD ≤ 5%.

**Table 1 marinedrugs-20-00479-t001:** The analysis of alginate-degraded zones produced by *Vibrio* sp. Ni1.

No.	Colony Diameter/cm	Alginate-Degraded Circle Diameter/cm	Ratio
1	1.13	4.58	4.05
2	1.02	4.82	4.72
3	0.86	4.62	5.37
Average value	1.00	4.67	4.67

**Table 2 marinedrugs-20-00479-t002:** The details of matched peptide sequences.

No.	Peptides Seq.	Candidate Protein ID	Length
1	NSITGHYWAVVK	AIY22661.1	12
2	AGVYNQFENGEAK	AIY22661.1	13
3	LLWEGDNKPVR	AIY22661.1	11
4	VVWEQER	WP_118120558.1	7
5	AGVYNQFENGEAK	WP_118120558.1	13
6	ADMGYGTSTENSSYIR	WP_118120558.1	16

**Table 3 marinedrugs-20-00479-t003:** The summary of some alginate lyases reported from *Vibrio*.

Protein Name	Molecular Mass (kDa)	Optimal pH/Temperature (°C)	Substrate Specificity	Enzymatic Product	Activating Cation	Reference
AlyA	67.4	8.5/25	alginate, polyM	DP1–4	Na^+^, Ca^2+^	[25]
AlyB	57.5	7.5/20–25	polyG	DP1–5	Na^+^, Ca^2+^	[25]
AlyA-OU02	65	8.0/30	polyM	DP2–4	Mg^2+^, Fe^2+^	[26]
alginate lyase	62.5	7.0/25	polyM	/	Zn^2+^	[27]
alginate lyase	60	7.0/40	polyMG	/	Na^+^, K^+^	[28]
Algb	55	8.0/30	polyMG,	DP2–5	Na^+^, Mg^2+^ Ca^2+^, Fe^2+^ Co^2+^	[29]
AlgNJ04	50	7.0/40	polyG	DP2–5	Na^+^, K^+^, Ca^2+^	[30]
Aly-IV	62	8.9/35	alginate	DP2–3	K^+^, Mg^2+^	[31]
Alg7A	56	7.0/30	polyMG, alginate	DP2–6	Ca^2+^, K^+^, Na^+^, Ba^2+^, Co^2+^, Mn^2+^	[32]
AlgB	67.7	8.0/35	polyM	DP2–3	Not found	This study

**Table 4 marinedrugs-20-00479-t004:** Primers used for cloning *alg*B.

Name	Sequence	Product
*alg*B-F0	GTGGTBTGGGAACARGAR	Partial *alg*B fragment
*alg*B-R0	YGGTTTRTTATCRCCYTCCCA	
SP1	CTCCACCCGTCCCATACCAAGAAT	Upstream flank sequences
SP2	GCACCTTCGCCTCGCCATTTTCAA	
SP3	AGTTGTCGAGGTCAGCTTCACCT	
P1	ACCCATTACTGCGGGTTGTTTGGGA	Downstream flank sequences
P2	AATGGCGAGGCGAAGGTGCAATTTA	
P3	AGACCCAGATTGTAGCACCAGTGATG	
*alg*B-QF	CGCGGATCCATGAATTCAGCAAAACTTATTTTGGT	Full *alg*B gene
*alg*B-QR	CCGCTCGAGCTACAGATGATTTACGTTCAAAGAGC	

Notes: the underline is restriction enzyme site.

## Data Availability

The data presented in this study are available on reasonable request from the corresponding author.

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
