# Peer review of "Cloning and Characterization of a Novel Endo-Type Metal-Independent Alginate Lyase from the Marine Bacteria *Vibrio* sp. Ni1"

_marinedrugs, 2022, doi:10.3390/md20080479_

Round 1

Reviewer 1 Report

The authors made an interesting work describing their work regarding the cloning and characterization of a novel endo-type metal-independent alginate lyase from the marine bacteria Vibrio sp. 3Ni1. Even if the presentation is accurate I considered it should be improved by following the next suggestions:

  1. In the abstract it is stated ORF, but it is not explained what does it stands for? Please explain the abbreviation and also the abbreviation of “1824bp”?
  2. In the introduction please consider to introduce recent reference regarding the alginate as polysaccharide https://doi.org/10.3390/pharmaceutics12100983
  3.  Figure 9 a and 11 should be more clear
  4. In line 358 to rewrite the “optimal” instead of the “optima”

Author Response

Response to the reviewer’s comments

  1. In the abstract it is stated ORF, but it is not explained what does it stands for? Please explain the abbreviation and also the abbreviation of “1824bp”?

Reply: The abbreviation of ORF and bp stands for open reading frame and base pair respectively.

  1. In the introduction please consider to introduce recent reference regarding the alginate as polysaccharide https://doi.org/10.3390/pharmaceutics12100983

Reply: Thanks for providing us a recent reference. This review has been added to reference [22], see line433.

  1.  Figure 9 a and 11 should be more clear 

Reply: Both Figure 9 a and 11 have been replaced with the corresponding higher resolution figures.

  1. In line 358 to rewrite the “optimal” instead of the “optima”

Reply: Revised as suggestion. (see line358)

Reviewer 2 Report

The manuscript content fits well with the journal scopes, and it needs to be revised.

1.      Provide introduction background information about important of marine-derived enzymes and marine microbial enzymes: See corresponding papers:

·        Alginate Lyases from Marine Bacteria: An Enzyme Ocean for Sustainable Future

·        Marine Microbial Fibrinolytic Enzymes: An Overview of Source, Production, Biochemical Properties and Thrombolytic Activity

·        Marine microbial L-glutaminase: from pharmaceutical to food industry

·        Marine Bacterial Esterases: Emerging Biocatalysts for Industrial Applications

·        Marine microbial alkaline protease: An efficient and essential tool for various industrial applications

2.      Please use the latest references for introduction section.

3.      Please correct some grammatical errors.

·        Line 61: Please correct sp. Ni1 have good to sp. Ni1 has a good

·        Line 86: analysis in NCBI database to analysis in the NCBI database

·        Line 93: has submitted to….. has been submitted to

·        Line 90: Vibrio alginalyticus to Vibrio alginolyticus

·        Line 108: rare that that to rare that

·        Line 115: Mutiple to Multiple

·        Line 117: the results

·        Line 117: to  with red box

·        Line 118: to  Domains

·        Line 118: to  boxes

·        Line 118: Funtional to Functional

·        Line 121: to labeled

·        Line 141: please change to AlgB sharply decreased when pH was lower than

·        Line 153: of AlgB was sharply to of AlgB has sharply

·        Line 190: to concentrations

·        Line 192: to Substrate

·        Line 196: to This illustrates that AlgB degrades

·        Line 198: these two alginate lyases are

·        Line 199: within them to decide the enzyme has polyM

·        Line 203: to substrate

·        Line 239: to used in a 100mL

·        Line 246: to as an expression host

·        Line 272: to in a vacuum

·        Line 273: in the dark at

·        Line 281: Thermo Fish Scientific to Thermo Fisher Scientific

·        Line 285: lyase

·        Line 286: degenerate

·        Line 330: to  ions served

·        Line 334: to average molecular

·        Line 349: to intermittently

·        Line 367: to  are in

4.      Please check the use of italics for microorganism names in lines 158, 200-202, 243, 245,

Author Response

  1. Provide introduction background information about important of marine-derived enzymes and marine microbial enzymes: See corresponding papers: 
  • Alginate Lyases from Marine Bacteria: An Enzyme Ocean for Sustainable Future
  • Marine Microbial Fibrinolytic Enzymes: An Overview of Source, Production, Biochemical Properties and Thrombolytic Activity
  • Marine microbial L-glutaminase: from pharmaceutical to food industry
  • Marine Bacterial Esterases: Emerging Biocatalysts for Industrial Applications
  • Marine microbial alkaline protease: An efficient and essential tool for various industrial applications
  1. Please use the latest references for introduction section.

Reply: Thanks for introduce these papers. We have carefully read them. The first review paper is close to the background and is referred and noted [15] at the reference. (See line415.)

  1. Please correct some grammatical errors.

Reply: All grammatical errors mentioned by the reviewer were revised.

4.Please check the use of italics for microorganism names in lines 157, 200-202, 243, 245

Reply: All microorganism names in the manuscript were check to italics form.

Reviewer 3 Report

This study is valuable for understanding alginate lyases. The authors cloned and characterized a novel alginate lyase from Vibrio sp. Ni1. Experimental methods and results are robust in support of the main conclusions. However, I have some suggestions for improvement. 

(1) At what point can we determine that this enzyme is new, and at what level of amino acid sequence homology can we determine that it is new? What makes it a superior property for industrial use compared to other enzymes? It was a little difficult to understand when I read it.

(2) The amino acid sequence of this enzyme suggests that it has two active sites, and both poly M and poly G are good substrates. The active residues of alginate lyase can also be estimated. More information can be obtained by creating mutants of each active site and analyzing them. As it stands, this paper is merely a report of the cloning of the enzyme, and it would be beneficial to the reader if some more data were added.

Others:

(1) Put a space between the number and the unit.

(2) Some genetic and specific names are italicized while others are not.

(2) Line 139: When specific activity is expressed in units of U/ml, it is difficult to compare with other enzymes.

Author Response

Response to the reviewer’s comments

(1)At what point can we determine that this enzyme is new, and at what level of amino acid sequence homology can we determine that it is new? What makes it a superior property for industrial use compared to other enzymes? It was a little difficult to understand when I read it.

Reply: The only one DNA sequence which shows high similarity to algB was identified by online BLAST analysis. This DNA sequence was from a genome of Vibrio penaeicida (NCBI accession number: AP025146.1). This algB-like gene hasn’t been reported. So we think this algB is a novel gene.

The AlgB can work well under the condition of pH 6-10 and 20-60 ℃ to adapt the industrial application with wide range of pH and temperature. Furthermore, AlgB can catalyze without any metal ion and is tolerant to NaCl.

(2) The amino acid sequence of this enzyme suggests that it has two active sites, and both poly M and poly G are good substrates. The active residues of alginate lyase can also be estimated. More information can be obtained by creating mutants of each active site and analyzing them. As it stands, this paper is merely a report of the cloning of the enzyme, and it would be beneficial to the reader if some more data were added.

Reply: Thanks for suggestions of further research. This submitted manuscript is focused on isolation and characterization of algB. We will do the following research to explore the catalytic mechanism.

Others:

(1)Put a space between the number and the unit.

Reply: Revised as suggestion.

  • Some genetic and specific names are italicized while others are not.

Reply: All genetic and specific names in the manuscript were revised to the italicized uniform.

(2) Line 139: When specific activity is expressed in units of U/ml, it is difficult to compare with other enzymes.

Reply: The used assay of alginate lyase activity is commonly applicated. According to the value of U/mL, the AlgB activity can be roughly compared with other reported enzymes.